## Effect of intensive treatment for schistosomiasis on immune responses to vaccines among rural Ugandan island adolescents: randomised controlled trial protocol A for the 'POPulation differences in VACcine responses' (POPVAC) programme

Gyaviira Nkurunungi [iD],[1] Ludoviko Zirimenya,[1] Jacent Nassuuna,[1] Agnes Natukunda,[1] Prossy N Kabuubi,[1] Emmanuel Niwagaba,[1] Gloria Oduru,[1] Grace Kabami,[1] Rebecca Amongin,[1] Alex Mutebe,[1] Milly Namutebi,[1] Christopher Zziwa,[1] Susan Amongi,[1] Caroline Ninsiima,[1] Caroline Onen,[1] Florence Akello,[1] Moses Sewankambo,[1] Samuel Kiwanuka,[1] Robert Kizindo,[1] James Kaweesa,[2] Stephen Cose,[1,3] Emily Webb,[4] Alison M Elliott,[1,3] on behalf of the POPVAC trial team

GN, LZ, JN and AN contributed equally.

► http://dx.doi.org/10.1136/bmjopen-2020-040425

For numbered affiliations see end of article.

**Correspondence to**
Dr Gyaviira Nkurunungi;
Gyaviira.Nkurunungi@mrcuganda.org

## ABSTRACT

**Introduction** Several licensed and investigational vaccines have lower efficacy, and induce impaired immune responses, in low-income versus high-income countries and in rural, versus urban, settings. Understanding these population differences is essential to optimising vaccine effectiveness in the tropics. We suggest that repeated exposure to and immunomodulation by chronic helminth infections partly explains population differences in vaccine response.

**Methods and analysis** We have designed an individually randomised, parallel group trial of intensive versus standard praziquantel (PZQ) intervention against schistosomiasis, to determine effects on vaccine response outcomes among school-going adolescents (9–17 years) from rural *Schistosoma mansoni*-endemic Ugandan islands. Vaccines to be studied comprise BCG on day 'zero'; yellow fever, oral typhoid and human papilloma virus (HPV) vaccines at week 4; and HPV and tetanus/diphtheria booster vaccine at week 28. The intensive arm will receive PZQ doses three times, each 2 weeks apart, before BCG immunisation, followed by a dose at week 8 and quarterly thereafter. The standard arm will receive PZQ at week 8 and 52. We expect to enrol 480 participants, with 80% infected with *S. mansoni* at the outset. Primary outcomes are BCG-specific interferon-γ ELISpot responses 8 weeks after BCG immunisation and for other vaccines, antibody responses to key vaccine antigens at 4 weeks after immunisation. Secondary analyses will determine the effects of intensive anthelminthic treatment on correlates of protective immunity, on waning of vaccine response, on priming versus boosting immunisations and on *S. mansoni* infection status and intensity. Exploratory

### Strengths and limitations of this study

► This will be the first adequately powered intervention study to investigate the effects of schistosomiasis treatment on vaccine responses in adolescents.

► Effects on both live-attenuated and inert vaccines will be studied.

► Our strong immunoepidemiological design and nested immunological studies will address specific hypotheses regarding pathways of effects.

► The sample archives developed will provide a major asset for exploration of new leads arising from this hypothesis-driven work, or for an alternative, 'systems biology' approach investigating (for example) transcriptome, microbiome and virome.

► Even with intensive anthelminthic intervention, it may be difficult to 'successfully' treat *Schistosoma* infection in our endemic setting due to re-infections; however, we still expect a substantial difference in intensity between the two trial arms.

immunology assays using archived samples will enable assessment of mechanistic links between helminths and vaccine responses.

**Ethics and dissemination** Ethics approval has been obtained from relevant ethics committes of Uganda and UK. Results will be shared with Uganda Ministry of Health, relevant district councils, community leaders and study participants. Further dissemination will be done through conference proceedings and publications.

**Trial registration number** ISRCTN60517191.

## INTRODUCTION

Vaccine-specific immune responses are often impaired, and vaccine efficacy and effectiveness lower, in tropical low-income countries compared with temperate high-income countries and in rural, compared with urban, LIC settings.[1–8] This has been recognised for both live vaccines (such as BCG,[2 3 5 9] polio[1] and yellow fever (YF)[4] vaccines) and non-live vaccines (such as influenza[10] and tetanus).[11] Investigational malaria[7] and viral-vectored tuberculosis[6] and Ebola[12] vaccines are also affected. Previous exposure to the target pathogen (or related organisms) may mask the benefit of the vaccine.[13 14] However, prevaccination exposure does not explain why Ebola trial vaccine-specific responses differ between healthy adults in UK and Senegal,[12] as the target organism is rare. Therefore, environmentally dependent mechanisms may play an important role.[5]

A long-held hypothesis is that parasites, particularly helminths, modulate vaccine responses through profound preimmunisation and postimmunisation bystander effects on immunological activation and regulation.[15–17] Helminths might also impact vaccine's responses through interactions with the complex ecosystem of mammalian gut bacteria, fungi, protozoa and viruses (the 'transkingdom' concept[18] detailed elsewhere in this journal (bmjopen-2020–040425)). Helminth-induced gut mucosa damage, the associated translocation of microbial products into the systemic circulation[19–21] and systemic immune activation or regulation mediated by microbial products might contribute to modulation of responses to vaccines and other infections.

Helminth-mediated modulation of vaccine responses has not been substantiated in human populations. No appropriately powered trials have been conducted to evaluate reversibility of their effects. In animal models, helminths generally impair priming and accelerate waning of vaccine responses, although effects vary with helminth species, vaccine type and the timing of infection and immunisation.[22] Most observational studies in humans also suggest suppressed or biased responses during helminth infection, especially during systemic infections, such as schistosomiasis and the filariases. There is modest evidence that treating geohelminths in humans improves responses to BCG[23 24] or oral cholera vaccine[25] and we found that schistosomiasis treatment improved the measles-booster response in preschool children.[26] There is, therefore, a strong case for a comprehensive assessment of the effects of helminths and their treatment on vaccine responses.

The extent to which helminths and related 'transkingdom' mediators causally and reversibly impact immunological characteristics associated with vaccine responses may best be determined by intervention studies. This trial protocol A of the 'Population differences in vaccine responses' programme (POPVAC A; Current Controlled Trials identifier: ISRCTN60517191) has been designed to evaluate the effect of *Schistosoma mansoni* and its treatment on vaccine responses. This study is one of three parallel trials whose designs and cross-cutting analyses are described separately in this journal (bmjopen-2020–040425, bmjopen-2020–040427 and bmjopen-2020–040430).

### Hypothesis

The overarching goal of the POPVAC programme is to understand population differences in vaccine responses in Uganda, in order to identify strategies through which vaccine effectiveness can be optimised for the low-income, tropical settings where they are especially needed. For this Trial A, we focus on the hypothesis that *S. mansoni* infection suppresses responses to unrelated vaccines; and that this effect can be reversed, at least in part, by intensive praziquantel (PZQ) treatment intervention.

### Objective

To determine whether there are reversible effects of chronic *S. mansoni* infection on vaccine response in adolescents, using an intervention study.

## METHODS AND ANALYSIS

### Setting and participants

Standard Protocol Items: Recommendations for Interventional Trials reporting guidelines[27] have been used. We will conduct an individually randomised, parallel group trial of intensive versus standard intervention against schistosomiasis (described below) in the *S. mansoni*-endemic Koome islands of Lake Victoria, Mukono district, Uganda.[28] We aim to enrol 480 participants, randomising 240 to each intervention arm. The study cohort will recruit participants aged 9–17 years in primary school years 1–6. Adolescents[29] in this study setting bear a heavy parasite burden.[30] In addition, this age group is a target group for vaccines against sexually transmitted infections (currently human papilloma virus (HPV); in future, it is hoped, for vaccines against HIV) and for booster immunisations.

### Recruitment criteria

#### Inclusion criteria

1. Attending the selected school and planning to continue to attend the school for the duration of the study.
2. Aged 9–17 years and enrolled in primary 1–6 (to avoid primary leaving examinations in late year 7, and loss to follow-up of children leaving after primary 7).
3. Written informed assent by participant and consent by parent or guardian.
4. Females agreeing to avoid pregnancy for the duration of the trial.
5. Willing to provide locator information and to be contacted during the course of the trial.
6. Able and willing (in the investigator's opinion) to comply with all the study requirements.

#### Exclusion criteria

1. Clinically significant history of immunodeficiency (including HIV), cancer, cardiovascular disease,

gastrointestinal disease, liver disease, renal disease, endocrine disorder and neurological illness.
2. History of serious psychiatric condition or disorder.
3. Moderate or severe acute illness characterised by any of the following symptoms: fever, impaired consciousness, convulsions, difficulty in breathing or vomiting; or as otherwise determined by the attending project clinician.
4. Concurrent oral or systemic steroid medication or the concurrent use of other immunosuppressive agents within 2 months prior to enrolment.
5. History of allergic reaction to immunisation or any allergy likely to be exacerbated by any component of the study vaccines, including egg or chicken proteins.
6. History of previous immunisation with YF, oral typhoid or HPV vaccine; previous immunisation with BCG or tetanus and diphtheria vaccine (Td) at age ≥5 years.
7. Tendency to develop keloid scars.
8. Haemoglobin less than 82 g/L.
9. Positive HIV serology.
10. Positive pregnancy test.
11. Female currently lactating, confirmed pregnancy or intention to become pregnant during the trial period.
12. Use of an investigational medicinal product or non-registered drug, live vaccine or medical device other than the study vaccines for 30 days prior to dosing with the study vaccine, or planned use during the study period.
13. Administration of immunoglobulins and/or any blood products within the 3 months preceding the planned trial immunisation date.

Further information on recruitment criteria can be found in online supplemental file 1.

## Interventions

We will individually randomise participants to intensive or standard PZQ treatment in a 1:1 ratio. The intensive arm will receive three doses of PZQ (approximately 40 mg/kg, assessed by height pole[31]) each 2 weeks apart prior to the first immunisation (the last of these 2–3 weeks before immunisation), followed by PZQ at 8 weeks (after primary endpoint sampling) and thereafter quarterly PZQ (approximately; timings adjusted to accommodate school terms) during follow-up. The standard arm will receive their first dose of PZQ at week 8 (after immunisation and after primary endpoint sampling) and a second dose at week 52 (to conform to Uganda Ministry of Health policy, which is annual treatment) (figure 1). No placebo will be used in this trial because all participants will be treated (although at different frequencies) and participants are unlikely to seek additional treatments outside the trial schedule: PZQ treatment is not popular because of the recognised (although temporary) adverse effects (described in online supplemental file 1).

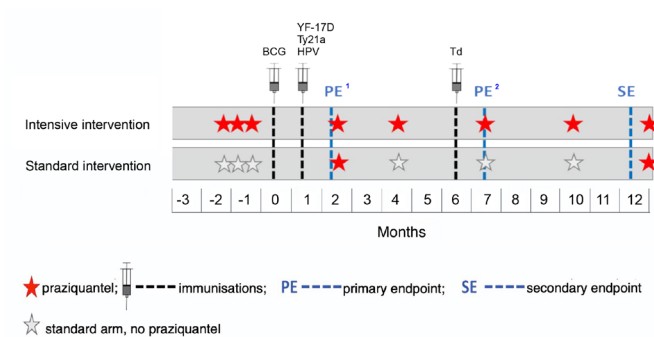

**Figure 1** Outline of immunisations and anthelminthic intervention.[1]Primary endpoints will be at 8 weeks post BCG and 4 weeks post yellow fever (YF-17D), oral typhoid (Ty21a), human papilloma virus (HPV) and tetanus/diptheria (Td) vaccination.[2]Primary endpoint for responses to Td given at 28 weeks.

## Randomisation and allocation to treatment arm

A randomisation code will be generated by an independent statistician using a randomly permuted block size (sizes 4, 6, 8 and 10) and used to allocate participants to either receive quarterly PZQ (intensive arm) or annual PZQ (standard arm). A set of envelopes will be prepared, labelled sequentially with the randomisation numbers and containing a card indicating the corresponding allocation (to intensive or standard treatment). The randomisation code will be kept securely by the trial statistician and made available only to those responsible for providing or preparing the trial interventions. A second copy will be held by a data manager or statistician not otherwise involved in the trial at the Medical Research Council/Uganda Virus Research Institute and London School of Hygiene and Tropical Medicine (MRC/UVRI and LSHTM) Uganda Research Unit. At enrolment, eligibility criteria will be checked and eligible participants will be allocated sequentially to the next randomisation number until the required sample size is achieved. Randomisation implementation will be done by a clinician using the sequentially numbered opaque sealed envelopes. When the next randomisation number in the sequence is allocated, the envelope bearing that number will be opened to reveal the allocation.

## Blinding

Clinicians and participants will not be blinded to the treatment allocation since they will not participate in outcome ascertainment; only immunology laboratory staff who are assessing trial outcomes will be blinded.

## Immunisations

We will study a portfolio of licensed vaccines (live and inert, oral and parental, priming and boosting) expected to be beneficial (in some cases, already given) to adolescents in Uganda. Our schedule (table 1 and online supplemental table S1) will comprise three main immunisation days (weeks 0, 4 and 28). Additional HPV immunisation will be provided for girls aged 14 years or above, and a

**Table 1** Immunisation schedule

| | Immunisation week 0 | Immunisation week 4 | Immunisation week 8 | Immunisation week 28 | Immunisation week 52 |
|---|---|---|---|---|---|
| Live vaccines | BCG vaccination/re-vaccination* | Yellow fever (YF-17D) Oral typhoid (Ty21a) | | | |
| Non-live vaccines | | HPV prime† | HPV boost for girls aged ≥14 years‡, § | HPV boost† and Td boost | Td boost¶ |

*Prior BCG status may vary (data on history and documentation of prior BCG, and presence of a BCG scar, will be documented although these approaches have limitations for determining BCG status).
†Both girls and boys will receive the HPV vaccine.
‡The National Expanded Programme on Immunisation recommends three doses of HPV vaccine for older girls.
§These doses will be given to comply with guidelines but outcomes specifically relating to these doses will not be assessed.
¶Priming by immunisation in infancy is assumed.
HPV, human papilloma virus; Td, tetanus/ diphtheria.

second Td boost will be given after completion of the study, to accord with the national Expanded Programme on Immunisation (EPI) routines, but the response to these will not specifically be assessed. Further rationale for the selection of vaccines is detailed in the online supplemental file 1.

### Schedule of immunisation and sampling
The schedule of immunisation and sampling is outlined in figure 1 and online supplemental table S1. Pre-immunisation vaccine responses will be assessed in baseline samples. While optimal timings for outcome measures vary between vaccines, sampling at 8 weeks post BCG and 4 weeks post YF-17D, Ty21a, HPV and Td is proposed for the primary endpoints, targeting the establishment of memory responses and approximate peak of antibody responses. A secondary endpoint at 1 year will assess waning. Immunisation postponement criteria are detailed in online supplemental file 1.

### Outcomes
#### Primary outcomes
These will be assessed in all participants.
1. BCG: BCG-specific interferon (IFN)-γ ELISpot response 8 weeks post BCG immunisation.
2. YF-17D: neutralising antibody titres (plaque-reduction neutralisation test) at 4 weeks post YF immunisation.
3. Ty21a: *Salmonella typhi* lipopolysaccharide-specific IgG concentration at 4 weeks post Ty21a immunisation.
4. HPV: IgG specific for L1-proteins of HPV-16/18 at 4 weeks post HPV priming immunisation.
5. Td: Tetanus and diphtheria toxoid-specific IgG concentration at 4 weeks post Td immunisation.

#### Secondary outcomes
These will be assessed in all participants and will further investigate estimates of protective immunity (for vaccines where these are available) and dynamics of the vaccine responses, as well as the impact of the interventions on parasite clearance.
1. Protective immunity. Proportions with protective neutralising antibody (YF); protective IgG levels (Tetanus

toxoid);[32] and seroconversion rates (Ty21a) at 4 weeks post the corresponding immunisation.
2. Response waning. Primary outcome measures (all vaccines) repeated at week 52, and area-under-the curve analyses. Parasitic infection may accelerate,[33] and anti-parasitic interventions delay, waning.
3. Priming versus boosting. Effects on priming versus boosting will be examined for HPV only, comparing outcomes 4 weeks after the first, and 4 weeks after the second vaccine dose.
4. Current *S. mansoni* infection status and intensity will be determined by serum/plasma levels of circulating anodic antigen (CAA). The method is quantitative, highly specific for *Schistosoma* infection, and much more sensitive than the conventional Kato Katz method.[34] CAA will be assessed retrospectively on stored samples collected at baseline, on immunisation days, and on primary and secondary endpoint days.

Furthermore, our sample collection will offer opportunities for an array of exploratory immunological evaluations on stored samples, focusing mainly on vaccine antigen-specific outcomes. Exploratory assays will provide further detail on the role of immunological profiles and transkingdom effects in mediating helminth modulation of vaccine-specific responses.

### Evaluation of parasite infection exposure
The following measures will also be assessed in all participants, and will be used to describe the general infection–exposure experience of the study participants.
1. Prior exposure to schistosomiasis will be evaluated by ELISA for IgG to schistosome egg antigen using stored blood samples collected at baseline.
2. The presence of other helminth infections will be determined retrospectively using stool PCR of samples collected at baseline and at weeks 28 and 52.[30] In accordance with national guidelines, all participants will be treated with albendazole or mebendazole after collection of samples for primary endpoints at weeks 8 and 28, and after collection of samples for secondary endpoints at week 52.

**Table 2** Power estimates (5% type 1 error rate for each primary outcome measure)

| | Difference in mean $\log_{10}$ transformed outcome between trial arms | | | | | | |
|---|---|---|---|---|---|---|---|
| Standard deviation ($\log_{10}$) | 0.08 | 0.10 | 0.12 | 0.14 | 0.16 | 0.18 | 0.20 |
| 192 intensive PZQ vs 192 standard PZQ (*Schistosoma mansoni* infected only) | | | | | | | |
| 0.3 | 65% | 83% | 94% | 98% | >99% | >99% | >99% |
| 0.4 | 42% | 59% | 75% | 87% | 94% | 98% | 99% |
| 0.5 | 29% | 42% | 56% | 69% | 80% | 88% | 94% |
| 0.6 | 21% | 31% | 42% | 53% | 65% | 75% | 83% |

Cells highlighted in grey correspond to >80% power; differences in mean $\log_{10}$ transformed outcome of 0.08, 0.10, 0.12, 0.14, 0.16, 0.18 and 0.20 are equivalent to geometric mean ratios for untransformed outcomes of 1.20, 1.26, 1.32, 1.38, 1.45, 1.51 and 1.59, respectively.
PZQ, praziquantel .

3. Current malaria infection status and intensity will be assessed retrospectively by PCR on stored samples collected on immunisation days and at week 52. Individuals presenting with fever will be investigated using rapid diagnostic tests for malaria and treated based on the results and according to prevailing national guidelines.
4. Prior malaria exposure will be evaluated by ELISA for IgG to malaria antigen using stored samples collected at baseline.

### Sample size considerations

Based on the literature[4 35 36] and preliminary data, we anticipate that, following $\log_{10}$ transformations that will be applied to normalise primary outcome measures, SDs of primary outcome measures will lie between 0.3 and 0.6 on this log scale, and that effective treatment may increase responses by approximately 0.2 on the log scale (based on Tweyongyere *et al.*[26] We have, therefore, powered our study to detect differences of this magnitude (0.2 on the log scale) or (in some cases) smaller (table 2). We assume *S. mansoni* prevalence of ≥80%.

Based on these assumptions and a two independent samples t-test, we plan to include 480 participants in total (240 quarterly PZQ and 240 annual PZQ); of whom 384 are expected to be *S. mansoni* infected,[28] giving 192 participants in each trial arm who are infected at baseline.

Table 2 shows power estimates, for 5% type 1 error rate for each primary outcome measure and assuming 20% loss to follow-up.

### Ethics and dissemination

Ethical approval has been granted from the Research Ethics Committee of the UVRI Regional Ethics Committee (REC; reference: GC/127/19/05/664) and the London School of Hygiene and Tropical Medicine (LSHTM, reference: 16032), and from the Uganda National Council for Science and Technology (UNCST, reference: HS2486) and the Uganda National Drug Authority (NDA, reference: CTA0093). Any protocol amendments will be submitted to ethics committees and regulatory bodies for approval before implementation.

Participants are adolescents and, therefore, a vulnerable human population. Care will be taken to provide adequate, age and education-status appropriate information and to ensure that it is understood; and to emphasise that participation is voluntary. Participants will be enrolled only when they have given their own assent and when consent has been given by the parent or guardian. Model consent and assent forms are shown in online supplemental file 2. No major risks to the participants are anticipated since all the treatments and vaccines to be given are licensed and known to be safe. The main risk to participants will be time lost from school work: we will work with teachers and parents to minimise disruption to classes, and will avoid enrolment of primary 7 students, since these classes are involved in national examinations. Further risks are discussed in online supplemental file 1.

Study findings will be published through open access peer-reviewed journals, presentations at local, national and international conferences and to the local community through community meetings. Anonymised participant level data sets generated will be available on request.

### Patient and public involvement

Concepts involved in this work have been discussed with colleagues at the Vector Control Division and EPI in the Ministry of Health (Uganda), with the Mukono District Council and with community leaders and village health teams from Koome subcounty. We also have held meetings to explain the proposed work to teachers, parents, participants and village members, and to address their questions about issues, such as study length, the study's ethical approval status, why adults were excluded from the study and to explain to them why boys will also receive the HPV vaccine. Study findings will be shared with these stakeholders and with participants.

### Data management and analysis

Sociodemographic information and clinical and laboratory measurements will be recorded and managed using Research Electronic Data Capture (REDCap) tools,[37 38] with paper-based forms as backup. All data will be recorded under a unique study identifier number. When paper forms must be used, data will be double

entered in a study-specific database, with standard checks for discrepancies. All data for analysis will be anonymised and stored on a secure and password-protected server, with access limited to essential research personnel.

Baseline characteristics including age, sex, school, location of birth, prior vaccination status, helminth infection and prior exposure status and malaria infection and prior exposure status will be summarised by trial arm. The effect of intensive (compared with standard) PZQ treatment on the outcomes will be analysed. Information on infection status will only be available after randomisation. The primary analysis will be done on individuals identified as infected at baseline (through randomisation, these will be balanced between treatment arms); this will test the hypothesis that treating the infection (and subsequent reinfections) reverses the parasite's effects on vaccine responses. If treating *S. mansoni* reverses adverse parasite effects on vaccine responses, this may be a beneficial public health intervention. However, routine screening for parasite infection before immunisation would be laborious. Secondary analyses will include all randomised individuals; this will provide insight into the broader benefit of the interventions as public health measures. The effect of intensive versus standard PZQ treatment on primary outcomes will be assessed using unpaired t-tests, with results presented as a mean difference in vaccine response measure together with 95% CI and p value. For all outcomes, we will investigate adjusting for corresponding baseline vaccine responses as this may improve the precision of effect estimates; this will be done using multivariable regression. We anticipate that outcomes will be positively skewed, and will apply log transformations to normalise distributions before analysis if required. The detailed analytical plan is available on the online trial registration site (http://www.isrctn.com/ISRCTN60517191).

## DISCUSSION

This will be the first adequately powered intervention study to investigate the effect of schistosomiasis treatment on vaccine responses in adolescents. This study will determine whether *S. mansoni* infection has a causal, reversible impact on the response to live or inert vaccines, including effects on vaccine replication, immune response profile, priming, boosting and waning. The results will add to understanding of population differences in vaccine responses and on interventions that may enhance responses. If treating helminths improves vaccine responses in adolescents, combined parasite-control/immunisation programmes offer an attractive, practical public health intervention for schools and communities.

There are risks associated with our approach to addressing the trial objective. First, there is a risk of failure to clear *S. mansoni* infections, and repeated reinfection during the trial. This issue can be challenging because of incomplete cure or maturation of immature worms after treatment, and lifestyles in endemic communities

that result in repeated exposure. To mitigate this, we will administer three PZQ treatments over a 6-week period before the first immunisations, and continuing quarterly treatment in the intensive arm. Second, there is a risk that *S. mansoni* infection has long-term effects, not removed by treatment, and mediated, for example, by epigenetic change.[39] However, studies show that parasite treatment results in immunological changes,[40 41] and our data suggest at least partial recovery of the measles vaccine response among young children treated for schistosomiasis.[26] By initiating intervention 6–8 weeks before the first immunisation, and providing repeated intervention in the intensive arms, we hope to achieve significant resolution of *S. mansoni* effects.

We are interested in the effects of removing *S. mansoni*. Treating parasites can induce acute immunological change due to release of previously hidden antigens.[42 43] To minimise such effects, immunisations will be given at least 2 weeks after PZQ (the longest practicable interval; figure 1).

Laboratory analyses will also highlight immune parameters and cellular populations that link environmental exposures to vaccine responses. Identifying processes associated with poor or good outcomes will inform strategies in vaccine design (for example, the genetic modification of vaccines, or innovative use of adjuvants to counter any adverse immunological milieu, currently an area of intense research for cancer vaccines[44]); ultimately supporting the development of effective vaccines tailored to the low-income settings that most need them.

### Study timeline

Applications for ethical approval were submitted in May 2018, with approval received in September 2018 (UVRI REC), May 2019 (NDA and UNCST), June 2019 (LSHTM). Collaborator/investigator/trial steering committee meetings were also held during the initial 12-month planning period. The study began recruitment in July 2019. Intervention will be up to 12 months, with completion of the project scheduled for September 2020.

**Author affiliations**
[1]Immunomodulation and Vaccines Programme, MRC/UVRI and LSHTM Uganda Research Unit, Entebbe, Uganda
[2]Vector Control Division, Republic of Uganda Ministry of Health, Kampala, Uganda
[3]Department of Clinical Research, London School of Hygiene and Tropical Medicine, London, UK
[4]MRC Tropical Epidemiology Group, Department of Infectious Disease Epidemiology, London School of Hygiene & Tropical Medicine, London, UK

**Correction notice** This article has been corrected since it first published. The provenance and peer review statement has been included.

**Acknowledgements** We thank the Uganda National Expanded Programme for Immunisation, Sanofi Pasteur and PaxVax for providing the human papilloma virus, yellow fever and oral typhoid vaccines, respectively. The BCG and tetanus/diphtheria vaccines were kind donations from the Serum Institute of India. We thank the Vector Control Division of the Ministry of Health and the Mukono district local government for their support. We also thank members of the Population differences in Vaccine response (POPVAC) programme steering committee (chaired by Professor

Richard Hayes) and the Data and Safety Monitoring Board (Dr David Meya, Professor Andrew Prendergast and Dr Elizabeth George).

**Collaborators** POPVAC trial teamPrincipal investigator: Alison Elliott. Project leader: Ludoviko Zirimenya. Laboratory staff: Gyaviira Nkurunungi, Stephen Cose, Rebecca Amongin, Beatrice Nassanga, Jacent Nassuuna, Irene Nambuya, Prossy Kabuubi, Emmanuel Niwagaba, Gloria Oduru and Grace Kabami. Statisticians and data managers: Emily Webb, Agnes Natukunda, Helen Akurut and Alex Mutebe. Clinicians: Anne Wajja, Milly Namutebi, Christopher Zziwa and Joel Serubanja. Nurses: Caroline Onen, Esther Nakazibwe, Josephine Tumusiime, Caroline Ninsiima, Susan Amongi and Florence Akello. Internal monitor: Mirriam Akello. Field workers: Robert Kizindo, Moses Sewankambo, Denis Nsubuga, Samuel Kiwanuka and Fred Kiwudhu. Boatman: David Abiriga. Administrative management: Moses Kizza and Samsi Nansukusa. Internal and external collaborators: Pontiano Kaleebu, Hermelijn Smits, Maria Yazdanbakhsh, Govert van Dam, Paul Corstjens, Sarah Staedke, Henry Luzze, James Kaweesa, Edridah Tukahebwa, Elly Tumushabe and Moses Muwanga.

**Contributors** AME conceived the study. AME, GN, EW, AN, JN, SC, LZ and JK contributed to study design. LZ, GO, PNK, EN, GK, RA, CN, CO, MN, CZ, SA and FA are site clinicians/nurses/clinical laboratory technicians providing valuable input on clinical considerations of the intervention. MS, SK and RK are field workers handling the organisational integration of the intervention. AN, AM and EW are involved in organisation of the databases, trial randomisation, treatment allocation and drawing up of analytical plans. GN, LZ, JN, AN, SC, EW and AME drafted the manuscript. All authors reviewed the manuscript, contributed to it and approved the final version.

**Funding** The Population differences in vaccine response programme of work is supported by the Medical Research Council (MRC) of UK (grant number MR/R02118X/1). SC and JN are supported in part by the Makerere University – Uganda Virus Research Institute Centre of Excellence for Infection and Immunity Research and Training (MUII-plus). MUII-plus is funded under the Developing Excellence in Leadership, Training and Science (DELTAS) Africa Initiative. The DELTAS Africa Initiative is an independent funding scheme of the African Academy of Sciences, Alliance for Accelerating Excellence in Science in Africa and supported by the New Partnership for Africa's Development Planning and Coordinating Agency with funding from the Wellcome Trust (grant 107743) and the UK Government. The MRC/Uganda Virus Research Institute and London School of Hygiene and Tropical Medicine (LSHTM) Uganda Research Unit is jointly funded by the UK MRC and the UK Department for International Development (DFID) under the MRC/DFID Concordat agreement and is also part of the European and Developing Countries Clinical Trials Partnership programme supported by the EU. The study sponsor (LSHTM) and funders had no role in study design; collection, management, analysis, and interpretation of data; writing of the protocol; and the decision to submit the protocol for publication.

**Competing interests** Alison Elliott reports a grant from the Medical research Council, UK (POPVAC programme funding). The rest of the authors declare that they have no conflicts of interest.

**Patient consent for publication** Not required.

**Provenance and peer review** Not commissioned; externally peer reviewed.

**ORCID iD**
Gyaviira Nkurunungi http://orcid.org/0000-0003-4062-9105

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
