## [Reviewer comments · BMJ Open]

ARTICLE DETAILS

TITLE (PROVISIONAL)	The effect of intensive treatment for schistosomiasis on immune responses to vaccines among rural Ugandan island adolescents: randomised controlled trial protocol A for the 'POPulation differences in VACcine responses' (POPVAC) programme
AUTHORS	Nkurunungi, Gyaviira; Zirimanya, Ludoviko; Nassuuna, Jacent; Natukunda, Agnes; Kabuubi, Prossy; Niwagaba, Emmanuel; Oduru, Gloria; Kabami, Grace; Amongin, Rebecca; Mutebe, Alex; Namutebi, Milly; Zziwa, Christopher; Amongi, Susan; Ninsiima, Caroline; Onen, Caroline; Akello, Florence; Sewankambo, Moses; Kiwanuka, Samuel; Kizindo, Robert; Kaweesa, James; Cose, Stephen; Webb, Emily; Elliott, Alison

VERSION 1 – REVIEW

REVIEWER	Umut Gazi Near East University Cyprus
REVIEW RETURNED	06-Jul-2020

GENERAL COMMENTS	It is a well written protocol that can be improved with minor additions. For instance, it would be better to include the vaccine and test names in the figure.
--

REVIEWER	Jose Ma. Angeles College of Public Health, University of the Philippines Manila, Philippines
REVIEW RETURNED	26-Jul-2020

GENERAL COMMENTS	The researchers aim to determine the effects of treating schistosomiasis with praziquantel to immune responses against a number of vaccines including BCG on day 'zero'; yellow fever, oral typhoid and Human Papilloma Virus (HPV) vaccines at week 4; and HPV and Tetanus/diphtheria booster vaccine at week 28. Although the study objectives are interesting, there are certain major points that the researchers need to address. Here are the major comments: (1) The main objective of the study is to determine the reversible effects of chronic Schistosoma mansoni infection on vaccine responses. How do you define and measure reversible effects? (2) The study will compare vaccine responses in 2 groups: participants receiving intensive dosage of PZQ and those who will receive standard dosage of PZQ. Why is there a need to do intensive and standard treatment? Is the main goal of the
--

intervention to fully treat the patient and not to measure the effectiveness of different treatment dosages against the parasite? The study groups set by the researchers are inappropriate to meet the objective of this study.

(3) Why is being positive to *S. mansoni* not part of the inclusion criteria for the participants? The study aims to determine the effects on vaccine responses of treating the schistosome worm and not of the drug intake. Although it was mentioned that the study target age group are the ones usually with heavy worm burden, an actual examination for detecting and quantifying *S. mansoni* among the participants should be done at the start of the study.

(4) What about other parasitic infections that might be present in the participants? How will the researchers address this problem that might also affect the vaccine responses?

Here are the minor comments:

(1) The manuscript needs English editing. Some terms used were abstract such as "well-powered".

(2) In the Introduction, there are several ideas/information given by the authors that need more explanation so the readers will understand, such as (a) UK-Senegal differences in Ebola trial, (b) environmental sensitization, (c) "trans-kingdom" mediators

(3) Line 22, 69. Vaccine efficacy is generally regarded as setting-independent. Are the investigators referring to vaccine effectiveness instead?

(4) Lines 25-26 The investigators may specify as to what exactly "repeated exposure to and immunomodulation by chronic helminth infections" are crucial to.

(5) Line 152 How do the investigators intend to elicit "intention to become pregnant" from prospective female study participants?

(6) Line 163 Why would a height pole be used to determine the participants' weight and thus compute drug dose?

(7) Line 286 Given the potential benefit as well of HPV vaccination to boys (e.g. anal cancer), investigators may otherwise simply explain the need for them to get one, rather than discuss with them whether to give it or not.

(8) Line 321 Have these schistosomiasis-induced epigenetic changes already been found to influence immune responses to vaccination?

(9) Line 328 Can investigators look further into previous studies that specify which "previously hidden antigens" are released with treatment that influence immune response to vaccination?

Supplementary Information

(10) Line 40 BCG has been proven to protect against extrapulmonary but not pulmonary TB infection.

(11) Line 104 Though specified in the main body of the paper, live attenuated vaccines are generally not recommended among immunodeficient individuals.

(12) Line 124 Would the investigators know if HIV testing is a voluntary or an opt-out program in Uganda?

	RECOMMENDATION: The manuscript be not accepted in its current form. Resubmission of the protocol might be done after revising based on the comments raised by the reviewer.
--	---

REVIEWER	Guangyu Tong Department of Biostatistics, Yale University
REVIEW RETURNED	01-Sep-2020

GENERAL COMMENTS	The data analytical plan is largely missing from the current protocol. The randomization process is not clearly described, either. This also, perhaps, leads to a disconnection between the study goal/purpose and the data analytical plan. Below are my more specific comments. (The page number needs to be fixed and it's now hard to make any specific references in the writing. I will refer to section names vaguely.)  1. In the section of Sample Size Calculation. The point "we anticipate that standard deviations (SDs) of 254 primary outcome measures will lie between 0.3 and 0.6 log10" is confusing. First, does this refer to standardized measures of outcomes? Second, I'm not aware of "log10" is a typical representation of the SD, or the authors are following some dosage response literature. Even if that is the case, 10 should not be written as a base for the logarithm. 2. Table 2 is quite informative on how the 80% power was achieved. However, the sd=0.6 row and the log difference = 0.2 column has 83% power, which suggests that the trial might be slightly over powered (compared to the typical 80% power). This should be a consideration if the enrollment of additional participants is not inexpensive. The authors need to clarify this. 3. The authors also want to clarify how their power was calculated, especially how the 20% loss to follow-up is accommodated in this calculation. For example, since the outcomes are collected after different times post treatment. Perhaps the loss of follow-up is a more severe issue for some later-measured outcomes but not for the earlier-measured ones? The authors need to clarify this. Also, if the intensive treatment group has potential benefits (e.g., more free shots seem to be beneficial), is there any possible non-compliance condition that participants assigned to the less intensive group want to move to the more intensive group? Relatedly, is there any plan for ITT analysis? 4. Some of the interventions are not for all participants. For example, HPV vaccine is only for girls aged 14+. This is not considered in the power analysis. 5. "We plan to include 480 participants in total (240 quarterly PZQ, 240 259 annual PZQ); of whom 384 are expected to be S. mansoni infected." Is there any basis for this reasoning, e.g., prevalence of disease on the site? The prevalence of disease seems to be really high in the study population. 6. There are apparently some covariates like gender and age mentioned here and there in the protocol, but they are not discussed in the data analysis plan. How covariates are used in the randomization process is not quite clear, either. The authors need to clearly state the covariates used in the randomization process as well as why these covariates are relevant and need to be balanced. Also, how are participants identified as infected at the baseline? Are
--

	those biomarkers also collected before the treatment is applied? 7. The current data analysis plan is over-simplified. The following issues need to be covered in the revision: statistical model used, covariate adjustment, handling of missing data, statistical software, etc, 8. Even though the randomization is at individual level, is there any concern about the potential clustering at the school level? If yes, this should also be considered in the data analytical plan. 9. Side note: The objective of the study is stated as “to determine whether there are reversible effects of chronic schistosoma mansoni infection on vaccine response in adolescents, using an intervention study.” By contrast, at the end of the data management and analysis: “this will test the hypothesis that treating the infection (and subsequent reinfections) removes the parasite’s effect. Secondary analyses will include all randomised individuals; this will provide insight into the potential benefit of the interventions as public health measures.” These statements of objective/hypotheses are quite disconnected, and a clear layout for them in the current protocol is missing. Indeed, the goal of this trial seems to be quite sophisticated, in that the biomarkers and the treatment may not necessary match each other. It seems that some indirect effect is being tested. The clarification of the study goal and how that is connected to the data analytical plan is very necessary.
--	--

REVIEWER	DR Irina Chis Ster St George's University of London
REVIEW RETURNED	04-Sep-2020

GENERAL COMMENTS	I have a positive view on this protocol. It is generally well written and it has a strong and valuable team behind. I do like the sensitivity analysis to the sample size and I appreciate the need of working with the log transform of the outcome as the assays distribution is likely to be skewed. The results are going to be interpreted through geometric mean ratios. May I ask the authors to complete the table 2 with numbers too? Perhaps underneath the power. I cannot comment on the epidemiological/clinical relevance of the difference as this is beyond my expertise. Secondly but more importantly, I would like to understand how the authors mitigate against the impact of the COVID 19 pandemic. How is the situation with the data collection? Has it stopped? Is it going to resume? It is not unusual that the recruitment starts to pick up later than it the official starting date. According to the paper it was supposed to have finished (September 2020). I would like the authors to comment on the implications of the pandemic on data collection including the possibility of losing power, how the recruitment and the follow up are holding and whether there is any possibility of simply transforming the experiment into an observational setting.
---

VERSION 1 – AUTHOR RESPONSE

COMMENTS FROM REVIEWER 1

Comment 1

It is a well written protocol that can be improved with minor additions. For instance, it would be better to include the vaccine and test names in the figure.

Response

The vaccine names are present in the Figure 1 legend, and have now also been added to the Figure.

Changes in the manuscript: Figure 1.

COMMENTS FROM REVIEWER 2

The researchers aim to determine the effects of treating schistosomiasis with praziquantel to immune responses against a number of vaccines including BCG on day 'zero'; yellow fever, oral typhoid and Human Papilloma Virus (HPV) vaccines at week 4; and HPV and Tetanus/diphtheria booster vaccine at week 28. Although the study objectives are interesting, there are certain major points that the researchers need to address.

Here are the major comments:

Comment 1

The main objective of the study is to determine the reversible effects of chronic *Schistosoma mansoni* infection on vaccine responses. How do you define and measure reversible effects?

Response

As stated in the Hypothesis section, we postulate that *S. mansoni* infection modulates responses to unrelated vaccines, and that these effects that can be "reversed", at least in part, by intensive *Schistosoma*-control intervention.

As reported in the Interventions section of the main text and in Supplementary Table S1, the intensive treatment arm will have received three doses of praziquantel each two weeks apart (approximately 40mg/kg) by the first immunisation (the last of these 2-3 weeks before immunisation) and by the time sample collection for measurement of primary outcomes (vaccine responses) is done. The standard arm will not have received any treatment by then. Because the study is in a high *S. mansoni* prevalence setting (Sanya et al., *Clinical infectious diseases*, 2019;68(10):1665-74. doi: 10.1093/cid/ciy761), the standard arm will provide the opportunity to assess the effect of *S. mansoni* infection on vaccine responses, providing a sample that can be used to infer responses before 'removal' of infection. The intensive arm will enable us to assess the effect of removal of active infection on vaccine responses and hence evaluate how 'reversible' the effect of *S. mansoni* infection on vaccine responses is. In the second paragraph of the discussion section we acknowledge, and discuss, risks associated with our approach to addressing the trial objective.

This study (POPVAC A, bmjopen-2020-040426) is one of three parallel trials whose designs and cross-cutting analyses will be described separately, but published concurrently, in this journal (POPVAC B, bmjopen-2020-040427; POPVAC C, bmjopen-2020-040430). Another manuscript describing cross-cutting analyses between all three trials is also under consideration by the journal (bmjopen-2020-040425). In the latter manuscript we describe, among several other things, comparisons to be conducted between participants of the urban-based POPVAC C and the rural based POPVAC A and B. POPVAC A will enable us to assess whether any differences in vaccine response between participants in the relatively urban POPVAC C and the rural POPVAC A are abrogated by intensive treatment in POPVAC A.

Changes in the manuscript: n/a

Comment 2

The study will compare vaccine responses in 2 groups: participants receiving intensive dosage of PZQ and those who will receive standard dosage of PZQ. Why is there a need to do intensive and standard treatment? Is the main goal of the intervention to fully treat the patient and not to measure the effectiveness of different treatment dosages against the parasite? The study groups set by the researchers are inappropriate to meet the objective of this study.

Response

The main goal of the study is to determine the effect (on vaccine responses) of removing active schistosome infection as fully as possible, which does entail an intervention that treats the patient as fully as possible. We know that single dose treatment does not clear all infections, hence the initial triple dose, hoping to optimise the possibility of clearing infection in the majority of subjects before immunisation. The “standard” arm will not have been treated at the primary end point at 8 weeks, hence allowing comparison between well treated, and untreated participants. CAA assays at both time points will tell us how well we have achieved the two desired schistosome infection status profiles.

We are also interested in longer term effects of reducing schistosome burden on vaccine responses and response waning but did not consider it ethical to leave children untreated for more than a year (given standard of care is annual treatment in schools in these communities). The “standard” arm is therefore a compromise from the “experimental” ideal.

Changes in the manuscript: n/a

Comment 3

Why is being positive to *S. mansoni* not part of the inclusion criteria for the participants? The study aims to determine the effects on vaccine responses of treating the schistosome worm and not of the drug intake. Although it was mentioned that the study target age group are the ones usually with heavy worm burden, an actual examination for detecting and quantifying *S. mansoni* among the participants should be done at the start of the study.

Response

The trial is taking place in a high *S. mansoni* prevalence setting (Sanya et al., *Clinical infectious diseases*, 2019;68(10):1665-74. doi: 10.1093/cid/ciy761); we expect prevalence of approximately 80% by measuring plasma levels of circulating anodic antigen (CAA). CAA will be assessed retrospectively on stored samples collected at baseline, on immunisation days, and on primary and secondary endpoint days (line 230-234).

Therefore, we expect to have enough individuals infected with *S. mansoni* at baseline, to be able to determine the effects on vaccine responses of treating the schistosome worm. As noted in the analysis section of the manuscript, “The primary analysis will be done on individuals identified as infected at baseline” (lines 302-303).

Changes in the manuscript: n/a

Comment 4

What about other parasitic infections that might be present in the participants? How will the researchers address this problem that might also affect the vaccine responses?

Response

We will use the PCR technique to assess stored stool for infection with geohelminths. Using stored blood, we will use the Modified Knott's method to assess infection with *Mansonella perstans* and PCR to detect malaria infection. Serology for prior malaria will also be conducted. This information is available in line 239-252 (main text) and in Supplementary Table S1.

Malaria and soil-transmitted helminths (STH) are less common than *S. mansoni* in our study settings: in our previous community-based trial (Sanya et al., *Clinical infectious diseases*, 2019;68(10):1665-74. doi: 10.1093/cid/ciy761) in the same setting we found prevalence of *P. falciparum* blood smear positivity of less than 4%, and hookworm prevalence at 11%.

Since this is a randomized trial, we anticipate that STH and malaria will be balanced between the trial arms. Nonetheless, this will be checked in the baseline data and adjusted for if a chance imbalance has occurred. STH and malaria will be considered as potential confounders, where appropriate, in observational comparisons, for example between the participants in this trial and those in related trials (POPVAC B; *bmjopen*-2020-040427 and POPVAC C; *bmjopen*-2020-040430). These comparisons are described in a companion manuscript (*bmjopen*-2020-040425) that is under consideration for simultaneous publication with the current one.

Changes in the manuscript: n/a

Here are the minor comments:

Comment 5

The manuscript needs English editing. Some terms used were abstract such as "well-powered".

Response

We use the term "well-powered" in the statistical sense, meaning that the proposed sample size is large enough to achieve a strong result. However, following the reviewer's comment, we have edited the main text, using the term "adequately powered" instead, where appropriate. We shall be happy to receive editorial suggestions that may improve the paper.

Changes in the manuscript: Lines 52, 87, 316

Comment 6

In the Introduction, there are several ideas/information given by the authors that need more explanation so the readers will understand, such as (a) UK-Senegal differences in Ebola trial, (b) environmental sensitization, (c) "trans-kingdom" mediators

Response

Statements in the introduction mentioning 'UK-Senegal differences' and 'environmental sensitization' have been edited to provide more clarity. Line 75-77 now reads: "However, pre-vaccination exposure does not explain why Ebola trial vaccine-specific responses differ between healthy UK and Senegalese adults, as the target organism is rare. Therefore, environmentally-dependent mechanisms may play an important role."

We have also edited line 82 to cite a related protocol manuscript (*bmjopen*-2020-040425) that explains "trans-kingdom" mediators in more detail. The above-named manuscript is under consideration for simultaneous publication with the current one, in *BMJ Open*.

Changes in the manuscript: line 75-77, Line 82

Comment 7

Line 22, 69. Vaccine efficacy is generally regarded as setting-independent. Are the investigators referring to vaccine effectiveness instead?

Response

Population/setting differences in both vaccine efficacy and effectiveness have been documented. Data are available from randomised controlled trials assessing reduction in disease incidence in vaccinated vs unvaccinated groups (efficacy trials), as well as less than optimally controlled studies assessing the ability of a vaccine to prevent disease outcomes (vaccine effectiveness). In the introduction we cite some of these studies. Examples include references 8 (efficacy) and 1 (effectiveness). Therefore, we have edited line 69 to mention both vaccine efficacy and effectiveness.

Changes in the manuscript: Line 69

Comment 8

Lines 25-26 The investigators may specify as to what exactly "repeated exposure to and immunomodulation by chronic helminth infections" are crucial to.

Response

We have edited line 26 in the text to provide more clarity.

Changes in the manuscript: Line 26

Comment 9

Line 152 How do the investigators intend to elicit "intention to become pregnant" from prospective female study participants?

Response

During the screening process, participants will be verbally asked their willingness not to become pregnant during the study period. Furthermore, females will also be tested for pregnancy at follow up visits before they receive any study related vaccination.

Changes in the manuscript: n/a

Comment 10

Line 163 Why would a height pole be used to determine the participants' weight and thus compute drug dose?

Response

This is standard practice in field distribution of praziquantel. We have provided a reference in line 163 that quotes the WHO-developed height pole system for administration of praziquantel.

Changes in the manuscript: Line 163

Comment 11

Line 286 Given the potential benefit as well of HPV vaccination to boys (e.g. anal cancer), investigators may otherwise simply explain the need for them to get one, rather than discuss with them whether to give it or not.

Response

We agree with the reviewer that HPV immunisation is also beneficial for boys, and we will include them in our studies. At national level in Uganda, HPV immunisation is not the policy at present, conceivably for reasons of vaccine cost. For our study, we have held meetings with teachers, parents,

participants and village members, to explain the need for boys to receive the HPV vaccine. We have rephrased line 291-292 to clarify this.

Changes in the manuscript: Line 291-292

Comment 12

Line 321 Have these schistosomiasis-induced epigenetic changes already been found to influence immune responses to vaccination?

Response

Long-term immunological effects of parasitic helminth infection have been demonstrated before in red grouse (Wenzel MA, Piertney SB. *Mol Ecol* 2014; 23:4256–73); we postulate that *S. mansoni* might also induce such effects. It is plausible that these could have bystander effect on vaccine response. We do not have evidence for this, but postulate that it is a risk.

Changes in the manuscript: n/a

Comment 13

Line 328 Can investigators look further into previous studies that specify which "previously hidden antigens" are released with treatment that influence immune response to vaccination?

Response

Release of previously cryptic antigens from dying worms has been postulated to account for part of the increase in immune responses after treatment in schistosomiasis ((1) M.E.J. Woolhouse & P. Hagan *Nature Medicine* volume 5, pages1225–1227(1999); (2) *J Infect Dis.* 2014 Jun 1; 209(11): 1792–1800; and (3). <https://doi.org/10.1086/344352>). It is plausible that these responses could influence vaccine responses. References 1 and 2 above have been added to the main text as references 43 and 44 (line 337).

Changes in the manuscript: Line 337

Supplementary Information

Comment 14

Line 40 BCG has been proven to protect against extrapulmonary but not pulmonary TB infection.

Response

BCG protects against pulmonary tuberculosis in some settings, but this is variable, with greater protection in populations further from the equator (e.g. Fine, *Lancet* 1995, cited as reference 2 in our protocol). Our POPVAC programme of trials partly seeks, as a key goal, to assess potential underlying causes for this.

Changes in the manuscript: n/a

Comment 15

Line 104 Though specified in the main body of the paper, live attenuated vaccines are generally not recommended among immunodeficient individuals.

Response

We agree with the reviewer. Indeed, we will exclude individuals with a clinically significant history of immunodeficiency (including HIV), cancer, cardiovascular disease, gastrointestinal disease, liver disease, renal disease, endocrine disorder and neurological illness. This information is available in the

exclusion criteria (line 134-136)

Changes in the manuscript: n/a

Comment 16

Line 124 Would the investigators know if HIV testing is a voluntary or an opt-out program in Uganda?

Response

Yes, HIV testing is a voluntary program in Uganda. Our trial participants have the option of refusing the HIV test, in which case they are not enrolled into the trial. Furthermore, positive HIV serology is an exclusion criterion.

Changes in the manuscript: n/a

COMMENTS FROM REVIEWER 3

Comment 1

The data analytical plan is largely missing from the current protocol. The randomization process is not clearly described, either. This also, perhaps, leads to a disconnection between the study goal/purpose and the data analytical plan. Below are my more specific comments. (The page number needs to be fixed and it's now hard to make any specific references in the writing. I will refer to section names vaguely.)

Response

The randomisation section (line 172) has been edited to make it clear that the randomisation code will be used to assign eligible participants to either receive quarterly PZQ (intensive arm) or annual PZQ (standard arm). Also, the permuted block sizes have been specified as 4, 6, 8 and 10. The data analytical plan is available on the online trial registration site (<http://www.isrctn.com/ISRCTN60517191>). We have included this information in the main text, as well as more details on the analyses to be conducted (line 309-314).

Changes in the manuscript: Line 172-184, Line 309-314

Comment 2

In the section of Sample Size Calculation. The point "we anticipate that standard deviations (SDs) of 254 primary outcome measures will lie between 0.3 and 0.6 log₁₀" is confusing. First, does this refer to standardized measures of outcomes? Second, I'm not aware of "log₁₀" is a typical representation of the SD, or the authors are following some dosage response literature. Even if that is the case, 10 should not be written as a base for the logarithm.

Response

The sample size calculation was based on parameters reported in literature (effect size and standard deviations) derived from vaccine responses that were log to base 10 transformed before analysis since we know these responses are usually highly skewed. Our data analysis plan specifies that we will apply log to base 10 transformations if our responses are similarly skewed. We have edited the section on Sample size considerations, and Table 2, to provide more clarity.

Changes in the manuscript: Line 254-259, Table 2.

Comment 3

Table 2 is quite informative on how the 80% power was achieved. However, the $sd=0.6$ row and the log difference = 0.2 column has 83% power, which suggests that the trial might be slightly overpowered (compared to the typical 80% power). This should be a consideration if the enrollment of additional participants is not inexpensive. The authors need to clarify this.

Response

We agree with the reviewer that the study is slightly overpowered, however recruitment of a few additional participants will be inexpensive considering recruitment and follow up is school based and there are direct benefits of receiving study vaccines for the participants involved. Also, with such power (if indeed the sd is 0.6) we will still be able to significantly detect a difference slightly lower than 0.2 which is still clinically relevant.

Changes in the manuscript: n/a

Comment 4

The authors also want to clarify how their power was calculated, especially how the 20% loss to follow-up is accommodated in this calculation. For example, since the outcomes are collected after different times post treatment. Perhaps the loss of follow-up is a more severe issue for some later-measured outcomes but not for the earlier-measured ones? The authors need to clarify this. Also, if the intensive treatment group has potential benefits (e.g., more free shots seem to be beneficial), is there any possible non-compliance condition that participants assigned to the less intensive group want to move to the more intensive group? Relatedly, is there any plan for ITT analysis?

Response

We assumed an overall loss to follow up of 20% covering a one-year period of follow up. The primary outcome will be collected at 8 weeks post BCG vaccination (~14 weeks post enrolment) and since recruitment and follow up is school based where the research team will visit and carry our study procedures at the schools, we don't expect more than 20% loss to follow up by the end of the one year planned follow up period. Therefore, loss to follow-up for outcomes assessed at 14 weeks post enrolment, may indeed be lower. Our study is not designed to allow participants to cross from one arm to another but it is possible that the participants in the standard arm could take PZQ outside study arrangements, however we don't anticipate this to be common. We plan to analyse the data by ITT and we shall also conduct sensitivity analysis (per protocol analysis) to determine how deviation from the protocol procedures affect our findings.

Changes in the manuscript: n/a

Comment 5

Some of the interventions are not for all participants. For example, HPV vaccine is only for girls aged 14+. This is not considered in the power analysis.

Response

Table 1 has been edited to clarify that the HPV vaccine will be given to both sexes, except for the week 8 timepoint, when it will be given only to girls aged 14+ that were not already vaccinated prior to this trial. This information is also available in the Supplementary information (line 80-92)

Changes in the manuscript: Table 1, Line 291-292.

Comment 6

“We plan to include 480 participants in total (240 quarterly PZQ, 240 259 annual PZQ); of whom 384 are expected to be *S. mansoni* infected.” Is there any basis for this reasoning, e.g., prevalence of disease on the site? The prevalence of disease seems to be really high in the study population.

Response

We assume *S. mansoni* prevalence of >80%, based on our previous studies in the study setting (Sanya et al., *Clinical infectious diseases*, 2019;68(10):1665-74. doi: 10.1093/cid/ciy761). We have included this reference to the text (line 261).

Changes in the manuscript: Line 261

Comment 7

There are apparently some covariates like gender and age mentioned here and there in the protocol, but they are not discussed in the data analysis plan. How covariates are used in the randomization process is not quite clear, either. The authors need to clearly state the covariates used in the randomization process as well as why these covariates are relevant and need to be balanced. Also, how are participants identified as infected at the baseline? Are those biomarkers also collected before the treatment is applied?

Response

The randomisation was not stratified by any covariates, i.e. simple permuted block randomisation was used as described; with our sample size of 480, we expect equal allocation of these covariates in each arm (and we will monitor this during recruitment). In addition, we shall conduct subgroup analysis by gender.

We expect prevalence of approximately 80%, based on our previous studies in the study setting (Sanya et al., *Clinical infectious diseases*, 2019;68(10):1665-74. doi: 10.1093/cid/ciy761). We will measure plasma levels of circulating anodic antigen (CAA) retrospectively using stored samples collected at baseline, on immunisation days, and on primary and secondary endpoint days (line 230-234). We expect to have enough individuals infected with *S. mansoni* at baseline, to be able to determine the effects on vaccine responses of treating the schistosome worm.

Changes in the manuscript: n/a

Comment 8

The current data analysis plan is over-simplified. The following issues need to be covered in the revision: statistical model used, covariate adjustment, handling of missing data, statistical software, etc,

Response

The detailed data analytical plan is available on the online trial registration site (<http://www.isrctn.com/ISRCTN60517191>). We have included this information in the main text and added an overview of the analysis approach for primary outcomes (line 308-314)

Changes in the manuscript: Line 308-314

Comment 9

Even though the randomization is at individual level, is there any concern about the potential

clustering at the school level? If yes, this should also be considered in the data analytical plan.

Response

We agree with the reviewer that there could be potential clustering at school level; however, we don't expect this clustering to affect our results since all the schools selected to be included in the study are located in the Islands of Lake Victoria with relatively similar exposure to helminth infection (proximity to the lake), similar diets etc. (our trial intervention is treatment of helminth infection before vaccination). However, a sensitivity analysis could be carried out to adjust for any clustering effect, and we will include this in our final analytical plan.

Changes in the manuscript: n/a

Comment 10

Side note: The objective of the study is stated as "to determine whether there are reversible effects of chronic schistosoma mansoni infection on vaccine response in adolescents, using an intervention study." By contrast, at the end of the data management and analysis: "this will test the hypothesis that treating the infection (and subsequent reinfections) removes the parasite's effect. Secondary analyses will include all randomised individuals; this will provide insight into the potential benefit of the interventions as public health measures." These statements of objective/hypotheses are quite disconnected, and a clear layout for them in the current protocol is missing. Indeed, the goal of this trial seems to be quite sophisticated, in that the biomarkers and the treatment may not necessary match each other. It seems that some indirect effect is being tested. The clarification of the study goal and how that is connected to the data analytical plan is very necessary.

Response

We have added more information to the Data management and analysis section to make the objective/hypotheses more coherent. The edited section now reads: "The primary analysis will be done on individuals identified as infected at baseline (through randomisation, these will be balanced between treatment arms); this will test the hypothesis that treating the infection (and subsequent reinfections) reverses the parasite's effects on vaccine responses. If treating *S. mansoni* reverses adverse parasite effects on vaccine responses, this may be a beneficial public health intervention. However, routine screening for parasite infection before immunisation would be laborious. Secondary analyses will include all randomised individuals; this will provide insight into the broader benefit of the interventions as public health measures."

It is true that treating the infection may have different effects on vaccine response compared to just never having the infection. We are well placed to assess whether this is the case, by comparing outcomes between the intensive treatment arm of this rural-based trial and another closely related trial set in a very low helminth exposure urban area (POPVAC C, bmjopen-2020-040430, protocol also under consideration for concurrent publication in *BMJ Open*). POPVAC C will not conduct anthelmintic treatment before the primary endpoint; its main goal is to assess the impact of BCG versus no BCG revaccination on vaccine responses among adolescents. The urban-based POPVAC C trial will further enable us to assess whether any differences between its participants and the rural POPVAC A participants are abrogated by intensive treatment in POPVAC A. These urban-rural comparisons are described in a protocol manuscript also under consideration for concurrent publication in *BMJ Open* (bmjopen-2020-040425).

Changes in the manuscript: Line 304-307

COMMENTS FROM REVIEWER 4

Comment 1

I have a positive view on this protocol. It is generally well written and it has a strong and valuable team behind.

I do like the sensitivity analysis to the sample size and I appreciate the need of working with the log transform of the outcome as the assays distribution is likely to be skewed. The results are going to be interpreted through geometric mean ratios. May I ask the authors to complete the table 2 with numbers too? Perhaps underneath the power. I cannot comment on the epidemiological/clinical relevance of the difference as this is beyond my expertise.

Response

We have edited the Table 2 legend to include the geometric mean ratio equivalent (on the original scale) of the difference that we are powered for on the log scale (additive scale).

Changes in the manuscript: Table 2.

Comment 2

Secondly but more importantly, I would like to understand how the authors mitigate against the impact of the COVID 19 pandemic. How is the situation with the data collection? Has it stopped? Is it going to resume? It is not unusual that the recruitment starts to pick up later than it the official starting date. According to the paper it was supposed to have finished (September 2020). I would like the authors to comment on the implications of the pandemic on data collection including the possibility of losing power, how the recruitment and the follow up are holding and whether there is any possibility of simply transforming the experiment into an observational setting.

Response

In line with the Ugandan Presidential Directives for COVID-19, the study was paused for three months. Fortunately, the recruitment process and the primary endpoint had been completed, and follow up of participants (up to week 28) was ongoing. Once research regulatory authorities allowed research to resume, we drafted a risk management plan which was approved by the institutional research ethics committee and Uganda National Drug Authority. With approval of national and local authorities, methods to ensure that participants are adequately followed up were put in place.

Changes in the manuscript: n/a

VERSION 2 – REVIEW

REVIEWER	Guangyu Tong Yale University
REVIEW RETURNED	13-Oct-2020

GENERAL COMMENTS	I have the following comments for the method and analysis. The main problem is the multiple testing and inconsistency in the analytical plan. 1. In the title of Table 2, " 5% significance level " mixed up the concepts of Type-1 error and hypothesis testing. It should be "5% type-1 error rate".2. The authors listed 5 primary outcomes. Usually, the power and sample size calculations should be tailored to all primary outcomes
--

	with Bonferroni corrections. But it is NOT considered in this protocol. The type-1 error rate needs to be adjusted with Bonferroni correction, which should be reduced to 1% for each outcome. For estimates, 99% CI should be used. This also means that the current power analysis is under-powered. If the authors want to avoid this issue, they should only pick one single primary outcome. 3. The power calculation part should also clarify the model/method used. It seems the power was based on two-sample t test based on the analysis. 4. The analysis part is improved, but still very brief. For example, on page 12, line 205, it says "All analyses will take baseline measurements into account". But this is missing in the "Data management and analysis" part. The "baseline characteristics" are analyzed separately, but not in the outcome model, which is a two sample t test. The authors also said "The detailed analytical plan is available on the online trial registration site (http://www.isrctn.com/ISRCTN60517191)." But in this link, it actually says "They will also add the statistical analysis plan to the trial registration before database lock". So, there are still no more details on this analysis. 5. It is still unclear what "baseline characteristics" are. There should be a part or a sentence that explicitly lists all your baseline characteristics, which is currently hard to track. The part of "Additional evaluation of parasite infection exposure" is confusing, as it is unclear if they are baseline covariates or outcomes or both. Will they be analyzed at all as outcomes? If not, why are they listed within the section of outcomes?
--	--

REVIEWER	Irina Chis Ster St George's University of London
REVIEW RETURNED	25-Oct-2020

GENERAL COMMENTS	This a well written and clearly explained protocol.
---

VERSION 2 – AUTHOR RESPONSE

REVIEWER 3

I have the following comments for the method and analysis. The main problem is the multiple testing and inconsistency in the analytical plan.

Comment 1

In the title of Table 2, " 5% significance level " mixed up the concepts of Type-1 error and hypothesis testing. It should be "5% type-1 error rate".

Response

We thank the reviewer for this correction. The phrase "significance level" has been changed to "Type-1 error rate" in both the Table 2 title and the text immediately above it.

Changes in the manuscript:

Title of Table 2, line 265

Comment 2

The authors listed 5 primary outcomes. Usually, the power and sample size calculations should be tailored to all primary outcomes with Bonferroni corrections. But it is NOT considered in this protocol. The type-1 error rate needs to be adjusted with Bonferroni correction, which should be reduced to 1% for each outcome. For estimates, 99% CI should be used. This also means that the current power analysis is under-powered. If the authors want to avoid this issue, they should only pick one single primary outcome.

Response

The aim of the POPVAC A trial is to assess, using a randomised design, the impact of removing schistosomiasis infections, on a portfolio of vaccines of different types and deliveries (e.g. live and inert, oral and parental). We do not know whether the impact of schistosomiasis and its removal will be the same for all vaccines, and are equally interested in all vaccines being studied. For this reason, specifying a single primary outcome would not be appropriate. Regarding the question of multiplicity, there are conflicting schools of thought on whether the type-1 error rate should be adjusted in the event of multiple primary outcomes being assessed, as summarised by Schulz and Grimes (Schulz KF, Grimes DA. *Lancet*. 2005 Apr 30-May 6;365(9470):1591-5. doi: 10.1016/S0140-6736(05)66461-6. PMID: 15866314). We have chosen not to make any formal adjustment for multiple testing in our analysis plan, predominantly for the following reasons: (1) this is an exploratory trial which aims to understand patterns of vaccine response and whether they may be impacted by schistosomiasis and its removal, it is not a confirmatory or regulatory trial and is not operating in a purely hypothesis-testing paradigm; (2) responses to different vaccines may well be correlated and thus making an adjustment such as a Bonferroni adjustment would most likely provide a severe overcorrection (as discussed by Sankoh et al. <https://doi.org/10.1002/sim.1557> ; *Stat Med* 2003; 22: 3133–50.) and would not be appropriate; (3) in future papers reporting trial findings, we will be cautious not to over-interpret findings, we will mention the possibility of any “significant” findings being due to chance (which is always a possibility regardless of whether type-1 error rate is further controlled) and we will interpret findings with respect to internal consistency (or inconsistency) in our results.

Changes in the manuscript: n/a

Comment 3

The power calculation part should also clarify the model/method used. It seems the power was based on two-sample t test based on the analysis.

Response

The method used for the power calculation has been specified as two independent samples t-test.

Changes in the manuscript: Line 262

Comment 4

The analysis part is improved, but still very brief. For example, on page 12, line 205, it says "All analyses will take baseline measurements into account". But this is missing in the "Data management and analysis" part. The "baseline characteristics" are analyzed separately, but not in the outcome model, which is a two sample t test. The authors also said "The detailed analytical plan is available on the online trial registration site (<https://clicktime.symantec.com/3DYvn2LU4uv3ckutCN3SQqv6H2?u=http%3A%2F%2Fwww.isrctn.com%2FISRCTN60517191>)."

But in this link, it actually says "They will also add the statistical analysis plan to the trial registration before database lock". So, there are still no more details on this analysis.

Response

To avoid confusion, the statement "All analyses will take baseline measurements into account" has been deleted from the "Schedule of immunisation and sampling" section and explained in the "Data management and analysis" section. The baseline measurements that we are particularly referring to here are pre-immunisation vaccine responses that will also be assessed at week 0 (i.e. before the POPVAC immunisations are given). This information has now been added to the "Schedule of immunisation and sampling" section (line 201-202). We plan to use a two independent samples t-test to compare responses between the study arms, however we shall investigate adjusting for vaccine responses measured at baseline (i.e. measured in samples collected before POPVAC A immunisations were provided) using multivariable regression. Although we expect baseline vaccine responses to be balanced between trial arms due to the randomisation, vaccine responses at baseline may be correlated with the corresponding trial outcome vaccine responses (measured after immunisations have been provided) and thus adjusting for them may improve the precision of the effect estimates. We do not plan to adjust for any other characteristics assessed at baseline as we anticipate that they will be balanced between trial arms due to the randomisation. This information is included in the detailed statistical analysis plan which is available on <http://www.isrctn.com/ISRCTN60517191> as an attachment in the section named "additional files" at the bottom of the page. The downloadable file is named "ISRCTN60517191_SAP_01Nov19.doc". This was uploaded to the ISRCTN trial registration record on 19th March 2020, as can be seen under the Editorial Note section and next to the downloadable file itself.

Changes in the manuscript:
Line 201-202, 316-318.

Comment 5

It is still unclear what "baseline characteristics" are. There should be a part or a sentence that explicitly lists all your baseline characteristics, which is currently hard to track. The part of "Additional evaluation of parasite infection exposure" is confusing, as it is unclear if they are baseline covariates or outcomes or both. Will they be analyzed at all as outcomes? If not, why are they listed within the section of outcomes?

Response

Baseline characteristics have now been specified as age, sex, school, location of birth, prior vaccination status, helminth infection and prior exposure status and malaria infection and prior exposure status (Line 303-304).

Thank you for pointing out the potential confusion regarding the "Additional evaluation of parasite infection exposure" section, which was included under the general Outcomes section. We have now included this as a separate section. These measures will be assessed in all participants, and will be used to describe the general infection-exposure experience of the study participants, and their distribution will be compared between trial arms (as listed in the baseline characteristics information above).

Changes in the manuscript:
Line 303-304, 239-241

VERSION 3 – REVIEW

REVIEWER	Guangyu Tong Yale University, United States
REVIEW RETURNED	07-Dec-2020
GENERAL COMMENTS	The revised draft shows much improvement in the statistical/analytical plan part. Still, one outstanding issue in the

	power calculation (on p.12/Table 2) is that it is still unclear with regards to which outcome variable this calculation is based upon. The authors need to clarify that. If the calculation is for all five primary outcomes, then the total type-1 error rate is 5%, but each variable will only share 1% (when evenly distributed).
--	---

VERSION 3 – AUTHOR RESPONSE

Comment 1

The revised draft shows much improvement in the statistical/analytical plan part. Still, one outstanding issue in the power calculation (on p.12/Table 2) is that it is still unclear with regards to which outcome variable this calculation is based upon. The authors need to clarify that. If the calculation is for all five primary outcomes, then the total type-1 error rate is 5%, but each variable will only share 1% (when evenly distributed).

Response

We thank the reviewer for highlighting this one outstanding point. To ensure clarity, we have updated the text immediately above Table 2 and the title of Table 2 itself to state that the type-1 error rate is 5% for each primary outcome measure. As explained in our previous response, we do not plan formal adjustment for multiplicity, and thus the power calculations as shown match with the planned analysis.

Changes in the manuscript:

Line 265, Table 2 title